# Learning brain dynamics for decoding and predicting individual differences

**Joyneel Misra**[1‡]*, **Srinivas Govinda Surampudi**[1‡], **Manasij Venkatesh**[1], **Chirag Limbachia**[2], **Joseph Jaja**[1], **Luiz Pessoa**[1,2]*

**1** Department of Electrical and Computer Engineering, University of Maryland, College Park, Maryland, United States of America, **2** Department of Psychology and Maryland Neuroimaging Center, University of Maryland, College Park, Maryland, United States of America

‡Authors with comparable contributions listed in random order.
* joyneelm@umd.edu (JM); pessoa@umd.edu (LP)

**Data Availability Statement:** The data and code used to train all models in this study are available in the following repository: https://github.com/LCE-UMD/GRU.

## Abstract

Insights from functional Magnetic Resonance Imaging (fMRI), as well as recordings of large numbers of neurons, reveal that many cognitive, emotional, and motor functions depend on the multivariate interactions of brain signals. To *decode* brain dynamics, we propose an architecture based on recurrent neural networks to uncover distributed spatiotemporal signatures. We demonstrate the potential of the approach using human fMRI data during movie-watching data and a continuous experimental paradigm. The model was able to learn spatiotemporal patterns that supported 15-way movie-clip classification (∼90%) at the level of brain regions, and binary classification of experimental conditions (∼60%) at the level of voxels. The model was also able to learn individual differences in measures of fluid intelligence and verbal IQ at levels comparable to that of existing techniques. We propose a dimensionality reduction approach that uncovers low-dimensional trajectories and captures essential informational (i.e., classification related) properties of brain dynamics. Finally, *saliency* maps and lesion analysis were employed to characterize brain-region/voxel importance, and uncovered how dynamic but consistent changes in fMRI activation influenced decoding performance. When applied at the level of voxels, our framework implements a dynamic version of multivariate pattern analysis. Our approach provides a framework for visualizing, analyzing, and discovering dynamic spatially distributed brain representations during naturalistic conditions.

## Author summary

Brain signals are inherently dynamic and evolve in both space and time as a function of cognitive or emotional task condition or mental state. To characterize brain dynamics, we employed an architecture based on recurrent neural networks, and applied it to functional magnetic resonance imaging data from humans watching movies or during continuous experimental conditions. The model learned spatiotemporal patterns that allowed it to correctly classify which clip a participant was watching based entirely on data from other participants; the model also learned a binary classification of experimental conditions at

**Funding:** L.P. is supported by the National Institute of Mental Health (R01 MH071589 and R01 MH112517) [https://grants.nih.gov/grants/funding/r01.htm]. M.V. was supported by a fellowship by the Brain and Behavior Initiative, University of Maryland, College Park [http://bbi.umd.edu/funding/main]. Movie watching data for this study were provided in part by the Human Connectome Project, WU-Minn Consortium (Principal Investigators: David Van Essen and Kamil Ugurbil; 1U54MH091657) funded by the 16 NIH Institutes and Centers that support the NIH Blueprint for Neuroscience Research; and by the McDonnell Center for Systems Neuroscience at Washington University. The funders had no role in study design, data collection and analysis, decision to publish, or preparation of the manuscript.

**Competing interests:** The authors have declared that no competing interests exist.

the level of voxels. We developed a dimensionality reduction approach that uncovered low-dimensional "trajectories" and captured essential information properties of brain dynamics. When applied at the level of voxels, our framework implements a dynamic version of multivariate pattern analysis. We believe our approach provides a powerful framework for visualizing, analyzing, and discovering dynamic spatially distributed brain representations during naturalistic conditions.

## Introduction

As brain data become increasingly *spatiotemporal*, there is a great need to develop methods that can effectively capture how information across space and time support behavior. In the context of functional Magnetic Resonance Imaging (fMRI), although data are acquired temporally, they are frequently treated in a relatively static manner (e.g., in event-related designs, responses to short trials are estimated with multiple regression assuming canonical hemodynamics). However, a fuller understanding of the mechanisms that support mental functions necessitates the characterization of *dynamic* properties, particularly during the investigation of more naturalistic paradigms, including movie watching [1], and other continuous paradigms [2, 3].

A central goal of neuroscience is to *decode* brain activity, namely to infer mental processes and processes based on brain signals. In fMRI, most approaches are spatially constrained and consider voxels in a local neighborhood ("searchlight") or across a few regions. Such models are useful to decode stimuli or mental state, particularly when the problem is well understood and localized anatomically [4–7]. Additional techniques decode brain activity based on whole-brain functional connectivity matrices [8], as well as machine learning methods [9]. The vast majority of decoding methods are static, that is, the inputs to classification are patterns of activation that are averaged across time ("snapshots") [10]. Some studies have proposed using temporal information in addition to spatial data [11–16]. In such cases, features used for classification are extended by considering a temporal data segment instead of, for example, the average signal during the acquisition period of interest. Another strategy to make use of temporal information capitalizes on dynamic functional connectivity, and how it is related to different conditions and individual differences [17–20].

Despite some progress in utilizing temporal information in decoding methods, key issues deserve to be further investigated:

- How can we characterize brain signals generated by dynamic stimuli in terms of generalizable (across participants) spatiotemporal patterns? How are such patterns distributed across both space and time? For example, although a recent study used recurrent neural networks to decode brain states from fMRI data, they investigated different working memory conditions (remembering faces, bodies, tools, or places) which are associated with stable brain states at the temporal scale of technique [21].

- Understanding the dimensionality of brain representations has become an important research question in recent years [22–24]. For example, for simple movements, neuronal signals across populations of neurons can be well approximated by a projection onto a two-dimensional representation [25]. This question is now starting to be addressed in the fMRI literature [26, 27]. Here, we tackle the problem of learning low-dimensional spatiotemporal signals from dynamic stimuli, including movie watching and continuous experimental paradigms.

- If spatiotemporal patterns capture important properties of brain dynamics, do they capture information about individual differences that are predictive of behavioral capabilities?

- Although multivariate techniques, including neural networks, can be successful at the level of prediction, *interpretability* is of key importance in the context of neuroscience. Therefore, here we quantified the relative importance of spatiotemporal features, specifically how brain regions contribute to input classification as a function of time.

- Can spatiotemporal decoding be applied at both the region-of-interest and voxel levels? Extending it to the voxel level allows investigators to probe finer spatial representations underlying task conditions or states of interest, and opens the way for dynamic representational similarity analysis [28].

In the present paper, we sought to address the above challenges by developing a *dynamic* computational framework built upon recurrent artificial neural networks. At the broadest level, our goal was to develop a *multivariate* and *temporal* technique to study fMRI signals that help address the following question: what combination of signals (from multiple regions or voxels), from particular temporal windows, support particular brain states and/or behavior? In essence, how can we characterize multivariate dynamics?

A central goal of our study was to investigate how *latent* representations obtained by learning brain dynamics captures useful information, despite being detached from the input signals. More broadly, our objective was to outline and validate an approach that can be employed when dynamic paradigms are employed. Finally, although we used non-linear neural networks, we investigated ways in which the approach can be used that go beyond using a "black box", including a combination of "importance" and lesion analysis.

## Results

Our goal was to develop a spatiotemporal decoding framework to predict a stimulus, mental state, behavior, and/or personality traits based on brain signals. A key element of our model was a recurrent neural network based on Gated Recurrent Units (GRUs; [34]) sensitive to current and past inputs (Fig 1A). To determine the generalizability of our findings, all results reported below were obtained with data not used in any fashion for training and/or tuning. For a complete description of the architecture, please refer to Materials and methods.

### Generalizability of spatiotemporal patterns

To test the system's ability to generalize spatiotemporal brain patterns across participants, we employed the *GRU classifier* to predict movie clip labels from functional MRI data available from the Human Connectome Project. We performed 15-way classification as a function of time (Fig 2A). Accuracy increased sharply during the first 60 seconds, and stabilized around 90 seconds. Mean classification accuracy after this transient period was 89.46% (Fig 2B). For completeness, a formal evaluation of chance performance based on the null distribution obtained through permutation testing resulted in a mean accuracy of 8.40% (1000 iterations; [54]).

### Is temporal information necessary for clip prediction?

If capturing temporal information and long-term dependencies are important for classification, performance should be affected by temporal order. To evaluate this, we shuffled the temporal order of fMRI time series for test data. To preserve the autocorrelation structure in fMRI

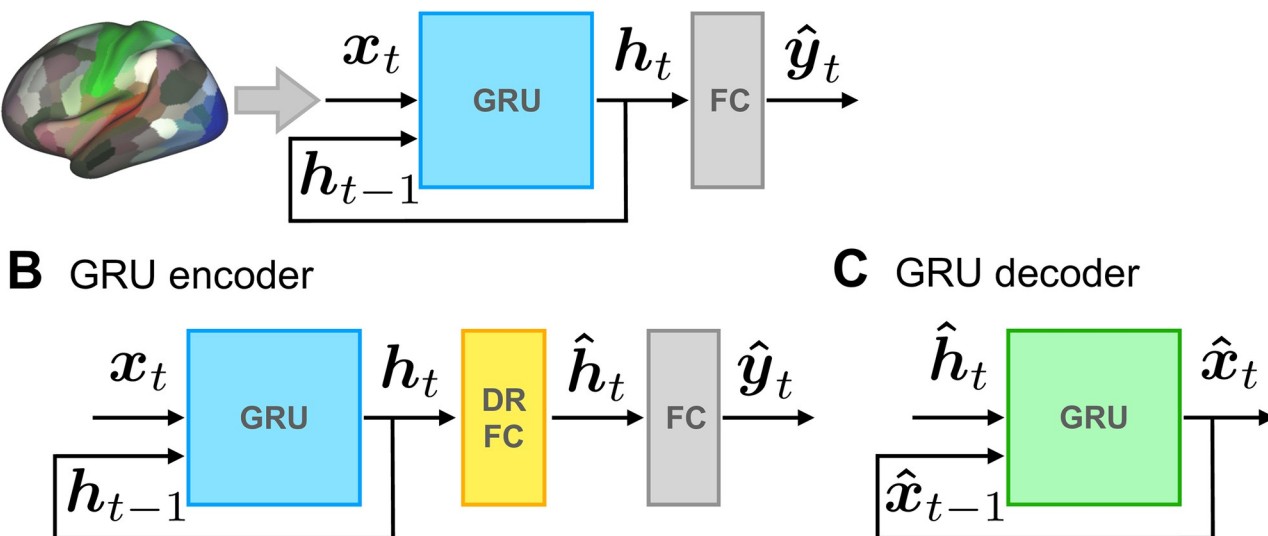

**Fig 1. Model architectures based on neural networks with gated recurrent units (GRUs).** (A) *Classifier*. At each time step, time series data,$x_t$, provided inputs. The recurrent neural network transformed the inputs into a latent representation, $h_t$, which then determined the output class scores, $\hat{y}_t$. The unit with highest activation determined the model's prediction of the input stimulus at each time. (B) *Dimensionality reduction*. The encoder shares the GRU component shown in A. GRU outputs, $h_t$, were first linearly projected to a lower-dimensional space using a fully-connected layer (DR-FC). Classification was then performed based on the low-dimensional representation, $\hat{h}_t$. (C) *Input signal reconstruction*. A separate GRU was trained independently to reconstruct the original brain signals based on the low-dimensional signals, $\hat{h}_t$.

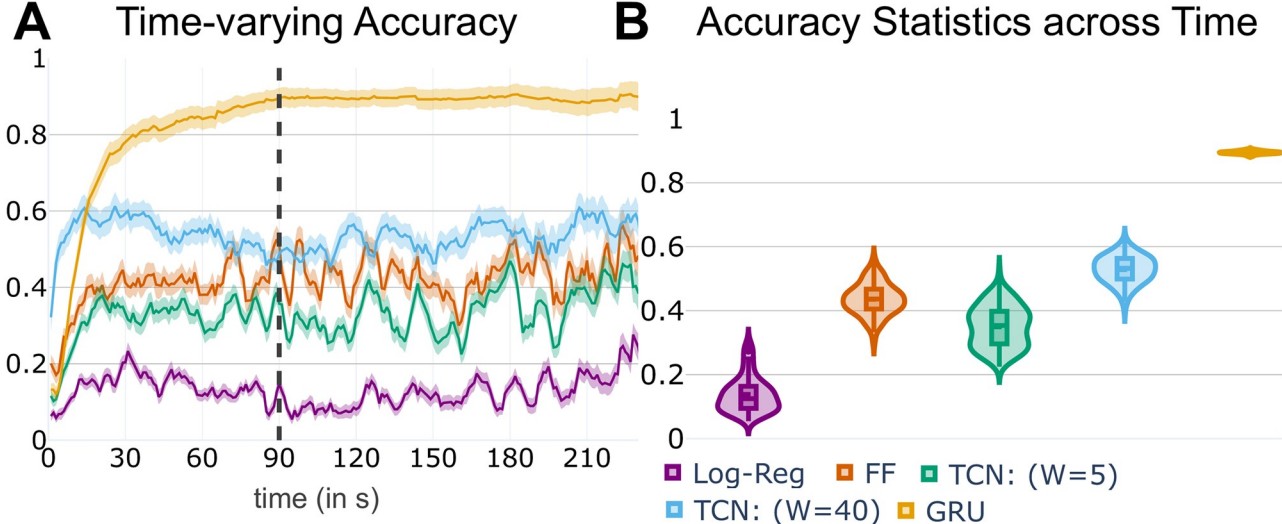

**Fig 2. Prediction of movie clip (15-way classification).** (A) Average clip prediction accuracy using the neural network architecture with gated recurrent units (GRUs) as a function of time (dark yellow). Accuracy increased sharply during the first 60 seconds, and stabilized around 90 seconds. Results using multinomial logistic regression (Log-Reg), a feed-forward architecture (FF; 1 layer, 103 units), temporal convolutional networks (TCN; kernel widths of 5 and 40) also also shown (see part B for labels). Error bars show the 95% confidence interval of the mean across test participants. (B) Summary of accuracy results after the 90-second transient period (see dashed line in part A).

data while shuffling, we used a wavestrapping approach [55]. Classification accuracy reduced considerably to 54.14%.

We then compared the GRU classifier to alternative schemes. Our objective was not to benchmark our proposal against possible competitors, but to further probe properties captured by our architecture. First, using a simple feed-forward (FF) network with one hidden layer, classification accuracy was 44.86%, which was considerably above the chance accuracy of 8.40% determined via permutation testing (the latter was essentially what we obtained when using multinomial logistic regression; 8.10%). We constrained our choice of units (103) such that the the total number of parameters of the feed-forward classifier (32, 563 parameters) was approximately equal to that of the GRU classifier (32, 559 parameters). As feed-forward classifiers do not capture temporal dependencies, we next employed temporal convolutional networks (TCNs). We used 25 kernels with a width ($W$) of 5 time steps such that the number of parameters of the classifier (37, 915 parameters) approximated that of the GRU classifier (see Materials and methods for kernel details). Classification accuracy was only 36.22%. Increasing the kernel width to 10 time steps (75, 415 parameters) modestly improved accuracy to 41.56%, and even using a kernel width of size 40 accuracy was only 54.85%, although the number of parameters (300, 415) was close to ten times that of the GRU classifier.

Together, the results above are consistent with the notion that temporal information is essential for high-accuracy classification, and that our GRU-based architecture is capable of capturing long-term dependencies that are missed by temporal convolutional networks, for example. Note, however, that temporal shuffling of the time series still allowed the GRU classifier to attain classification at around 54% correct. As the temporal shuffling was unique for each training permutation, order information was broken down, indicating that non-temporal information supported considerably above-chance performance. It is conceivable that spatial information of specific movie scenes provided sufficient information to account for that level of accuracy.

## Low-dimensional trajectories as spatiotemporal signatures

We investigated the dimensionality of the inputs required for successful stimulus prediction by using a supervised non-linear dimensionality reduction approach (Fig 1B). The original dimensionality of the input, $x_t$, was the number of regions ($N_x$ = 300). After learning, the GRU outputs provide a low-dimensional latent spatiotemporal representation of the input, which is encoded in the vector $h_t$ ($N_h$ = 32). To further reduce dimensions, GRU signals, $h_t$, were linearly projected onto a lower-dimensional space, $\hat{h}_t$, using a fully connected layer. We refer to this weight matrix as the *Dimensionality Reduction Fully-Connected (DRFC)* layer, and to this model as the *GRU encoder*. Since $\hat{h}_t$ is a low-dimensional representation of the history of $x_t$, the inputs are not treated independently, effectively leading to non-linear temporal dimensionality reduction. As in the original classifier, a final FC layer was used to predict labels based on $\hat{h}_t$.

Performance with low-dimensional encoding was surprisingly high; 3 dimensions yielded 71.21% accuracy. In Fig 3A, we show low-dimensional signals for each clip, which we refer to as *trajectories*: consecutive low-dimensional $\hat{h}_t$ states, $(\hat{h}_1(t), \hat{h}_2(t), \hat{h}_3(t))$, describe the trajectory in *state space*. In other words, at each time $t$, the value of each unit is plotted along the ($x$, $y$, $z$) axes.

To quantify a notion of proximity between trajectories, we computed the Euclidean distance between them as a function of time (Fig 3B). To compute the distance between clips $A$ and a reference clip $B$, we first computed the mean trajectory of $B$ averaged across participants. Then, for each participant's clip $A$ trajectory, we computed the Euclidean distance between $A$

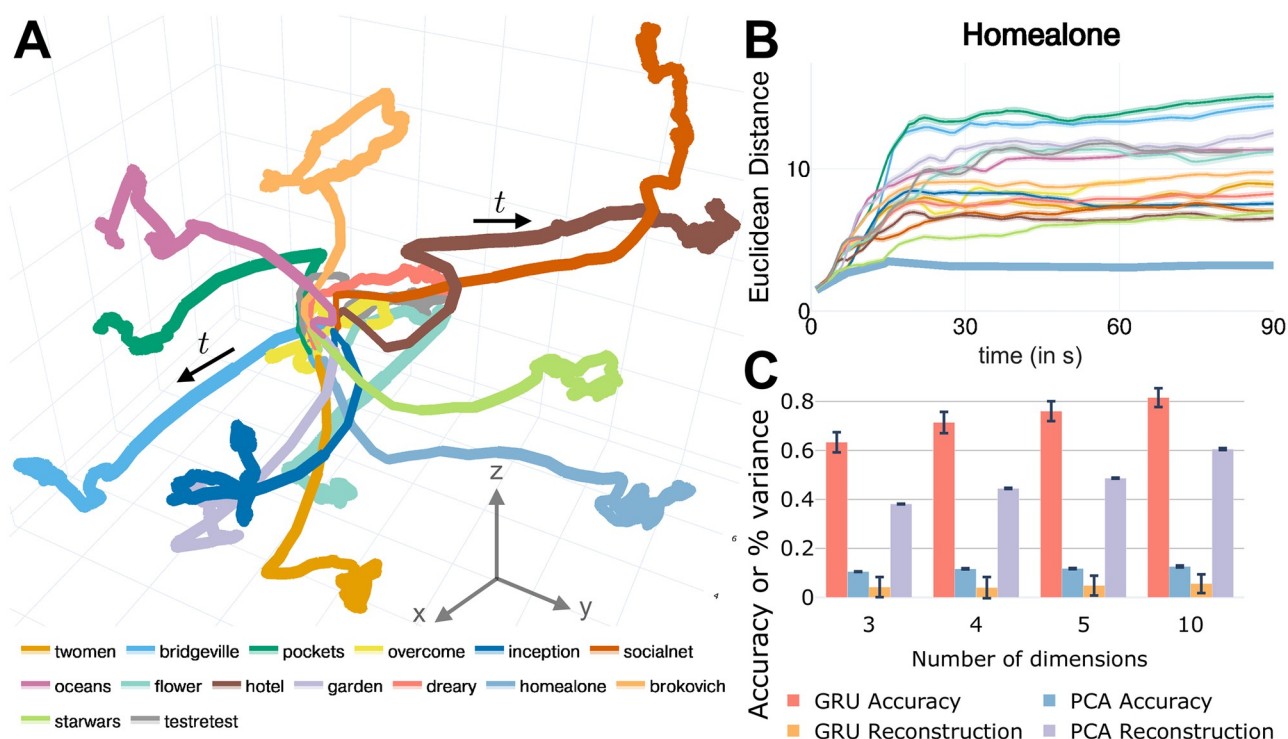

**Fig 3. Low-dimensional trajectories.** (A) Trajectories for all clips. Solid line: mean trajectory averaged across participants (line thickness is scaled by variance, which was highest at the end of the clip). All trajectories progressed away from the center (see sample arrows). The inset provides clip abbreviations. (B) Euclidean distance between trajectories. The Euclidean distance between the clip trajectory while watching *Home Alone* and the mean trajectory across participants for a second clip was computed. The thicker line corresponds to the distance of participants' *Home Alone* trajectories to the mean of this clip. The same results are shown for all clips in S1 Fig. (C) Clip prediction accuracy and fraction of variance captured after reconstruction using low-dimensional models. Error bars correspond to the standard error of the mean across participants.

and the mean trajectory of *B* at every time step. Accordingly, the proximity of a clip with itself was not zero (indicated by a thicker line), and reflected the variability of participant trajectories around the clip's average. As expected, the evolution of low-dimensional trajectories closely reflected the temporal accuracy obtained using the original GRU classifier. Clip trajectories were initially close to one another, but slowly separated during the first 60- 90 seconds of the clip. The results in Fig 3B also help clarify why the trajectories in (Fig 3A) originate from a common part of space. In the first few seconds, discrimination is very low and all trajectories are very close to one another in three dimensions. As time progresses, discrimination increases and the trajectories gradually separate from one another.

We further investigated low-dimensional projections with 4, 5, and 10 dimensions, at which point performance (87.84%) was not far (within 2%) from that of full dimensionality (Fig 3C). These results reveal that latent representations with as few as 10 dimensions captured essential discriminative information. The effectiveness of the GRU encoder in capturing low-dimensional information can be appreciated by applying principal components analysis on the input data, $x_t$, which yielded very low prediction accuracy, although it was capable of recovering 60% of the input signal variance with 10 dimensions (Fig 3C).

To investigate the content of the latent space uncovered by the GRU encoder, we performed the following analysis. By construction, the low-dimensional vector captures information to classify the input signals. How much information about the input does this projection preserve? To evaluate this, we considered the low-dimensional signal, $\hat{h}_t$, and used it to try to

reconstruct the input signal, $x_t$. Within our framework, it is natural to do so using a GRU decoder (Fig 1C), which reconstructs the input time series (the reconstructed signal is called $\hat{x}_t$). Doing so captured a very modest amount of signal variance (less than 10%).

## Spatiotemporal importance maps

The GRU representation that supports classification lies in a latent space that is fairly disconnected from the original brain activation signals, $x_t$. It is therefore important to relate GRU states to brain activation. We defined a *saliency* measure to capture the contribution of a brain region to classification as a function of time (see Materials and methods). This analysis attempts to identify sets of brain regions, and particular temporal windows, that are important for the task at hand.

Fig 4A shows a saliency map at $t = 8$ for the *Star Wars* clip, also shown dynamically for the first 60 seconds (S2 Video). The saliency time series in a few brain regions shows that values were relatively high initially and gradually decreased over the first minute of the clip (Fig 4B). The evolution of saliency for the *Home Alone* clip was somewhat similar (see also the saliency map video for the *Brokovich* clip in S1 Video).

## Lesion analysis

We further investigated region importance by performing a lesion analysis. After training, first we lesioned each of the seven large-scale networks described by Schaefer and colleagues [32] (Fig 5A). Lesioning the `visual`, `somato-motor`, and `default` networks produced the largest drop in test accuracy. Lesion of the other networks produced essentially no deficits.

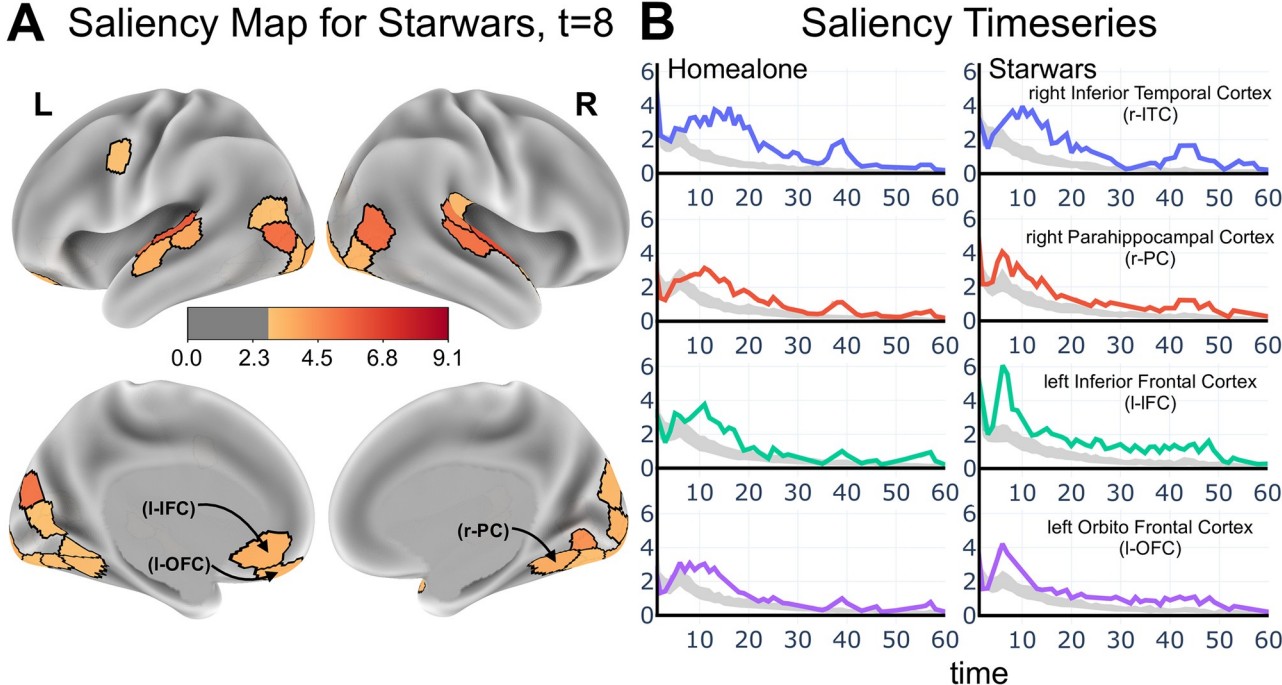

**Fig 4. Determining brain contributions to classification.** (A) Saliency map for the "Star Wars" clip. For illustration, the top 30 regions are shown. Color scale in arbitrary units. See also the video for dynamics. (B) Saliency values for the first 60 seconds of two clips. The gray bands corresponds to the 95-th percentile of null saliency values generated via permutation testing. Scale of the *y*-axis is arbitrary.

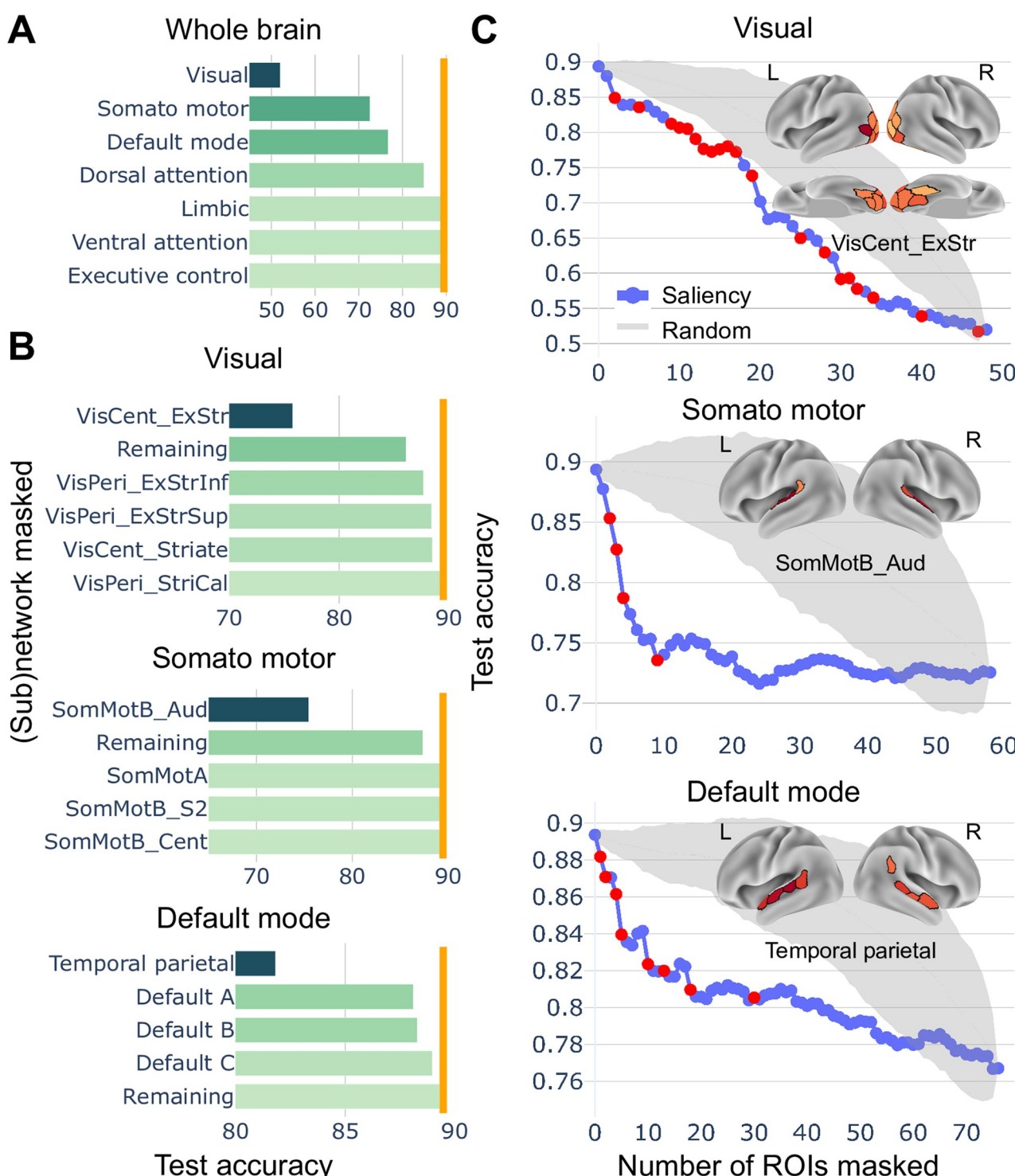

**Fig 5. Lesion analysis.** (A) The impact of a lesion in each of the seven standard networks is shown. The orange line shows accuracy without any lesion. (B) The impact of lesions to specific subnetworks was also evaluated, revealing greater contributions to clip classification by specific sets of brain regions. (C) Comparison of removal of regions from a given network (`visual`, `somato-motor`, or `default mode`) relative to a random set of regions from that same network. Removing regions in descending order of *overall* saliency (see text) impacted classification more than selecting regions without considering saliency (without replacement). For illustration, we also display the regions of the most impactful subnetworks observed in part B (red dots; also shown in the insets). The gray region shows the 95% confidence interval when the same number of regions were excluded at random.

Next, we capitalized on the parcellation available in terms of seventeen networks, allowing us to investigate how sectors of the seven "canonical" networks contributed to performance(Fig 5B). We found that the `temporal-parietal` was the only one of the `default` network subgroups that had an appreciable impact on performance. In addition, only the `central extra-striate` regions in the `visual` network, and the `early auditory` regions in the `somato-motor` network had more appreciable impacts.

Both saliency and lesion analyses highlight brain regions that are important for classification. To relate the two, we tested the following conjecture: regions with higher saliency values should have larger effects on classification when lesioned. To test this, we sequentially excluded regions progressing from the highest to the lowest *overall* saliency values, which was defined as the mean saliency value across time, clips, and participants. We performed tests on the `visual`, `somato-motor`, and `default` networks, which were found to be important for classification. According to the conjecture, accuracy should more sharply decrease after excluding the most salient ROIs. To test for the alternate hypothesis, we lesioned ROIs based on random selection (repeated for 1000 iterations). Thus, we contrasted performance when selecting regions of the, say, `visual` network based on saliency values relative to selecting regions from the same network randomly. Fig 5C shows that when ROIs were lesioned in sequence of decreasing saliency there was a larger drop in accuracy compared to random selection, therefore supporting our conjecture.

## Predicting behavior

Recent studies have employed brain data to predict participants' behavioral capabilities or personality-based measures [56–60]. We tested the extent to which spatiotemporal information captures individual-based measures based on our recurrent neural network framework. Here, we targeted individual scores for fluid intelligence and verbal IQ. The same approach used for classification was employed, with the model predicting a scalar, namely, a participant's score. We used all clips for training (rather than training a separate model based on each clip) to promote learning representations that are not idiosyncratic to a particular clip.

Fig 6 shows model performance temporally for the *Star Wars* clip, one of the best predicting clips. At every time point, predicted scores are correlated with actual participants' scores. For fluid intelligence, the observed correlations were often above chance levels (gray zone indicates 95% confidence level at specific time $t$), but more modestly for verbal IQ. As our model made predictions as a function of time, we developed a permutation-based test to evaluate the predictions while controlling for multiple comparisons (fluid intelligence, $p = 0.0013$; verbal IQ: $p = 0.0038$). The supplementary figures (S2, S3, S4 and S5 Figs) show results for all clips. As an alternative to evaluating prediction at every time point, we developed an "oracle test" that uses training data to learn to guess a time to test prediction once on the separate test set (See Materials and methods). For reference, we compared our approach to connectome-based predictive modeling based on patterns of functional connectivity (CPM; [51]), possibly the state-of-the-art in this regard (see also [56, 58]). We include values predicted with this technique to provide an informal comparison, as this method provides a single prediction per clip, unlike our approach which is time varying.

## Dynamic multivariate pattern analysis

Thus far, we applied our model to region-based activation patterns. The approach can also be employed to perform voxel-based (or grayordinate-based) dynamic multivariate pattern analysis (MVPA). Here, we applied our model to the prediction of experimental conditions from a dataset collected in our laboratory [3]. Briefly, in the "moving circles" paradigm, over periods

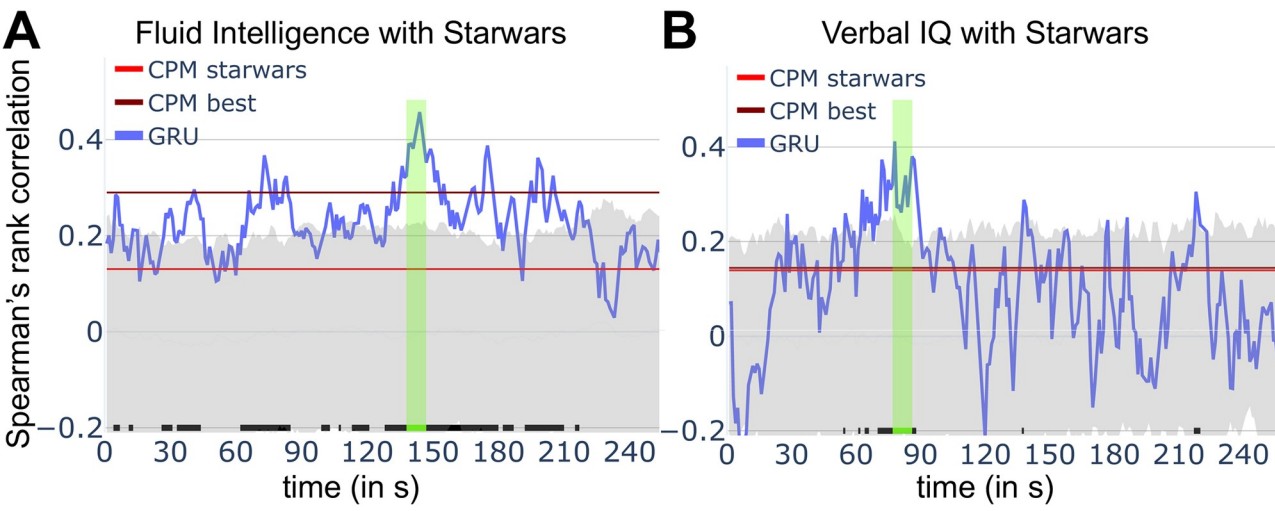

**Fig 6. Prediction of behavior.** (A) Prediction of fluid intelligence scores as a function of time. Prediction (blue) fluctuates considerably but consistently exceeds chance values (indicated by the tic marks at the bottom). Values obtained by connectome-based predictive modeling (CPM) are indicated for comparison (red: CPM applied to *Star Wars* data; maroon: highest value applying CPM across all clips). The gray region indicates the 95th percentile region based on permutation testing (N.B.: applies to our method only, not CPM). (B) Prediction of verbal IQ as a function of time. Only short periods of time of the Star Wars clip exceeded chance levels. The green bar indicates a segment of the time series that is significant at the 0.05 level corrected for multiple comparisons.

of three-minute blocks, two circles moved on the screen, at times approaching and at times retreating from each other. If they touched, participants received a non-painful but highly unpleasant electrical shock. The movement of the circles was smooth but not predictable. In particular, the circle motions were set up to include multiple instances of "near misses": periods of approach followed by retreat; in such instances, the circles came close to each other and retreated just before colliding. Based on near-misses, we defined *approach* and *retreat* states: each lasted seven time steps (total of 11.25 seconds) during which the circles approached or retreated from each other.

We performed dynamic MVPA by using voxels from the anterior insula, a region strongly engaged by the experimental paradigm [2, 3]. Classification accuracy ranged from 59- 63% over the approach and retreat segments (the upper bound of the 95% confidence interval determined with permutation testing was 50.37%). We also computed saliency in a voxelwise manner; values were substantially higher at the outset of a approach/retreat segment and relatively lower for the subsequent time points. Fig 7 shows a snapshot at $t = 6$ seconds, illustrating the spatial organization of the map.

## Discussion

We sought to characterize distributed spatiotemporal patterns of functional MRI data during movie watching and continuous task conditions. To do so, we employed recurrent neural networks sensitive to the long-term history of the input signals, and applied them to the problem of input classification. The model was tested on brain data both at the level of region of interest and voxels. All neural network training and model tuning were performed using a training set independent of the test set to establish the model's ability to generalize learned representations to unseen participants. The framework we developed is general and its subcomponents can be easily exchanged to utilize other algorithms (e.g., long short-term memory networks, or novel developments).

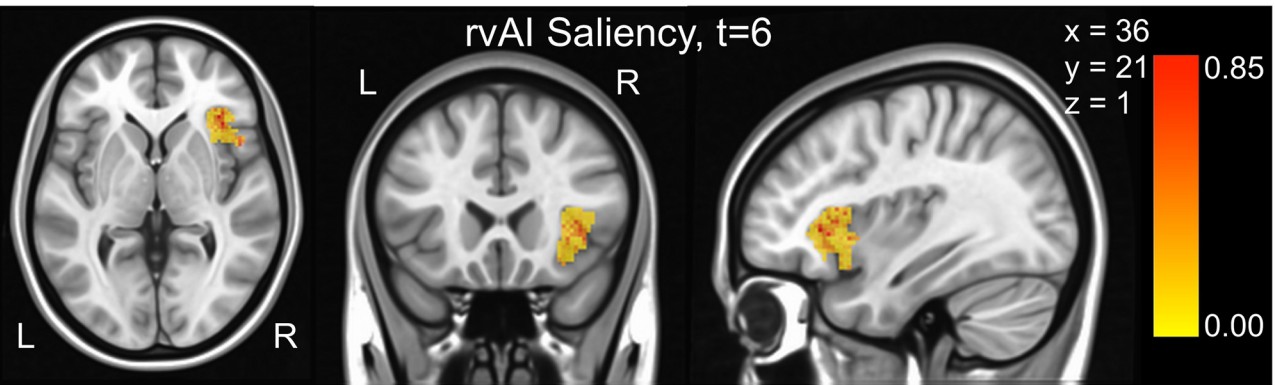

**Fig 7. Voxelwise prediction of experimental conditions: Threat vs. safe.** Only voxels from the anterior insula were employed. Saliency values at $t = 6$ are shown; for illustration no thresholding was applied.

## Spatiotemporal representations

Spatiotemporal information learned in a given set of participants generalized very well to new participants, where challenging 15-way classification was ~90% correct for movie clip data. These results are noteworthy because they suggest that activation patterns distributed across space and time are shared across participants when viewing naturalistic stimuli. To understand the type of information captured by our model, we compared it to a few simple schemes, including a simple feedforward network, and temporal convolutional networks. The low performance with these models (~40%) suggests that long-term dependencies are essential for input classification. Indeed, temporal shuffling of the input time series decreased performance to ~50%. As temporal shuffling preserves the input signals but scrambles their sequence, these results reveal that the precise temporal order is key for the ability to generalize to unseen data.

## Latent dimensionality

The question of the "inherent" dimensionality of brain signals has attracted considerable attention in recent years [22, 23, 61–64]. Here, we investigated this issue for the movie clip data, where inputs were 300-dimensional based on the ROI time series. As the number of units in the hidden layer was 32, a considerable amount of dimensionality reduction was accomplished at the outset. Further investigation revealed that this signal could be further reduced to ~10 dimensions without compromising prediction accuracy appreciably. Strikingly, even with three dimensions, accuracy was relatively high (77.30%).

Two issues about dimensionality deserve attention here. First, as the transformation of the input by the model is non-linear, when stating that one has, say, a 10-dimensional space, the non-linearity of the space must be kept in mind, as contrasted to a space obtained by linear techniques, such as those based on singular value decomposition. Second, the system processes temporal data. Thus, in a sense the dimensionality of the system is considerably higher (for example, "10d+time").

Whereas the mapping onto a lower dimensionality space preserved classification information about the input category, it was very poor at reconstructing the input time series itself. Thus, the content of the latent low-dimensional representation was substantially distinct from the fMRI signal itself. This observation raises the concern that the non-linear projection of the original input signal is too far removed from the brain time series. In other words, that the system works in a strongly "black box" fashion. In the present case, we believe this concern is

considerably assuaged by the results of the saliency and lesion analysis (see below). Another way to probe this issue would involve applying the present architecture to datasets with a more clear input or behavioral "space". For example, using the model to learn about the spatiotemporal content of classes of movies, much like research with static stimuli organized along certain natural axes [28], allowing the representational content of the low-dimensional signals to be evaluated. Finally, note that related issues apply to linear dimensionality reduction, too. In many cases, the first principle component may carry high signal variance, but other components associated with low, and even very low, input signal variance explain most of the behavioral variance [65–68].

## Saliency and lesion analysis

Although prediction is valuable in some settings (e.g., clinical diagnosis), our general goal was to develop a framework to characterize distributed spatiotemporal brain signals. Thus, one of our objectives was to identify sets of brain regions, and particular temporal windows, that are important for the classification task at hand, having in mind future applications of our model to characterizing spatiotemporal patterns linked to behavioral conditions or states.

Together, the saliency and lesion analyses uncovered several notable findings. Regions of the of the visual and auditory cortex were important for classification. Whereas this was to be expected, it serves as an important confirmation that the modeling approach captured biologically relevant information. Corroborating this statement, the model also identified as important regions of the inferior temporal cortex (such as the fusiform gyrus and the parahippocampal cortex), which are involved in high-level object recognition (not simply "low-level" visual processing).

Our findings also illustrate how the method can be used to formulate hypotheses that can be subsequently studied in more targeted studies. We found, for example, that the inferior frontal cortex and the orbitofrontal cortex supported clip classification, too. The lesion analysis also identified specific parts of the default network that are particularly important. Thus, the model makes novel predictions that can be further studied, for instance, by developing clips to address the types of spatiotemporal information being captured by these regions (social? emotional?). Taken together, our results support the idea that the model captured biologically relevant spatiotemporal information that can inform specific research questions.

## Individual differences

Recent work has sought to predict individual differences based on functional MRI data [51, 56–60, 69, 70]. Our model was capable of predicting fluid intelligence and verbal IQ from unseen participants at levels comparable to, and possibly better than, established methods such as connectome-based predictive modeling, which is based on information from functional connectivity matrices [51]. Some movies (e.g., *Star Wars*, *Social Net*, *Oceans*) performed much better at prediction than others, and at particular segments of the movie [71].

Why does movie watching correlate with individual differences, including fluid intelligence? We note that the main reason for our analysis was to demonstrate the feasibility of our approach, not a priori hypotheses. The approach adopted here holds more promise when the content of the naturalistic task is chosen or designed to investigate the behavioral dimension of interest. For example, Finn and colleagues found that trait paranoia is linked to brain signals during ambiguous social narratives ([72]; for further discussion, see also [71]). We plan to apply the method described here to investigate individual differences in reward sensitivity and anxiety-related measures in work employing dynamic paradigms [3].

Should prediction of behavioral differences be performed in a time-varying manner? Such strategy could be potentially beneficial if particular segments were better "tuned" to particular individual differences; for example, very suspenseful parts might be particularly associated with anxiety-related traits. This possibility remains speculative and needs to be investigated in future studies. Nevertheless, we showed that it is possible to perform above-chance prediction by making a single prediction of behavior. In this approach, the best-predictive segment is selected based on training data, and this specific temporal window is used with unseen, test data.

### Dynamic multivariate pattern analysis

We applied the model developed here with inputs at the level of voxels, so as to implement dynamic MVPA. We tested the approach in a dataset involving continuous changes in threat level based on the proximity of two moving circles. Voxels from the anterior insula were used to predict stimulus condition (threat vs. safe). Although accuracy was relatively modest (∼60%), the results demonstrate the feasibility of the approach. Saliency maps also uncovered subsectors of the anterior insula with increased importance for prediction, illustrating how it is possible to identify voxels that contribute to decoding the experimental conditions, opening an avenue for the application of representational similarity analysis [28] in a dynamic setting.

### Other approaches

Decoding approaches using some temporal information have been used in the past [11–16]. Recurrent neural networks have also been employed for brain decoding more recently, albeit in some cases on relatively static task paradigms, such as working memory which generates stable brain states at the temporal scale of fMRI [21, 73]; working memory conditions can be predicted even when temporal information is eliminated [68]. Because most fMRI datasets are typically small, they are often evaluated using cross-validation without assessing their generalization on held-out data (see Discussion by [74]). The size of the datasets studied allowed us to test the model in a separate set of participants thereby evaluating the model's ability to generalize beyond trained data (see also [73]). Finally, we stress that the goal of the present study was not to describe a model that outcompetes others but to describe a general, modular modeling approach to investigating spatiotemporal dynamics of brain data, here applied to fMRI.

## Materials and methods

### Data

**Movie data.**   We employed Human Connectome Project (HCP; [29]) data of participants scanned while watching movie excerpts (Hollywood and independent films) and other short videos, which we call "clips". The legend of Fig 3 provides further information. All 15 clips contained some degree of social content. Participants viewed each clip once, except for the *test-retest* clip that was viewed 4 times. Clip lengths varied from 65 to 260 seconds (average: 198 seconds).

We employed all available movie-watching data, except those of 8 participants with runs missing; thus we used $N$ = 176 participants. Data were sampled every 1 second. The preprocessed HCP data included FIX-denoising, motion correction, and surface registration [30, 31]. We analyzed data at the region of interest (ROI) level, with one time series per ROI (average time series across locations). We employed a standard 300-ROI cortical parcellation [32]. Thus, the input time series consisted of a vector of $N_x$ = 300 dimensions at each time, $t$.

Individual ROI time courses were normalized by z-scoring (i.e. centered to zero mean and unit standard deviation) to help remove potential differences across runs/participants.

**Moving-circles paradigm.**   We employed data from a separate dataset collected in our laboratory [3]. Briefly, the experimental paradigm was as follows. Over a period of 3 minutes, two circles moved on the screen, at times approaching and at times retreating from each other. If they touched, participants received a non-painful, but highly unpleasant electrical shock. The movement of the circles was smooth but not predictable. In particular, the circle motions were set up with multiple instances of what we called "near misses": periods of approach followed by retreat; in such instances, the circles came close to each other and retreated just before colliding. Based on near-misses, we defined *approach* and *retreat* states: each lasted 6–9 seconds during which the circles approached or retreated from each other.

Data from $N = 122$ participants were employed (sampled every 1.25 seconds). Preprocessing steps included motion correction (ICA-AROMA via FSL). We analyzed data at the voxel level from the right anterior insula, which corresponds to region numbers 230 and 231 of the `Schaefer-300` 17-network parcellation [32]. The input time series consisted of a vector of $N_x = 745$ dimensions at each time, $t$.

## Performance evaluation

**Accuracy.**   At each point in time, $t$, we computed accuracy as the average true positive rate (TPR) across inputs of a given class (movie clip or experimental condition). Specifically, for each class label $c$:

$$\text{TPR}_c = \frac{\sum_{n=1}^{N_c} \text{I}\{\text{predicted\_label}_n = c\}}{N_c} \tag{1}$$

where $N_c$ is the total number of data samples belonging to class $c$, predicted_label$_n$ is the label predicted by the model for the $n^{\text{th}}$ data sample, and I is an indicator function which equates to 1 if predicted_label$_n = c$ and 0 otherwise. We computed accuracy at each time step and visualized it as a function of time (see Fig 2).

**Training/testing split and cross-validation.**   Movie data from 100 participants were used for training, and the remaining 76 participants were used as a test set (thus completely invisible to model tuning/learning). There were more participants in the train set to have a little more data for training. To determine optimal hyperparameters for clip prediction, we employed a 5-fold cross-validation approach on the training set. In each fold, participants in the training set were not included in the validation set. After determining optimal hyperparameters, we retrained the model on the entire training data. Final results were generated exclusively based on test data. A similar approach was used for the the moving-circles paradigm, where data from 62 and 60 participants were used as a training set and test set, respectively. For cross validation, the folds had size {13, 13, 12, 12, 12}.

## Basic recurrent architecture for classification

Recurrent neural networks allow signals at the current time step to be influenced by long-term past information. Although it was originally difficult to develop effective learning procedures for these algorithms (e.g., vanishing/exploding gradients prevented them from learning relationships beyond 5–10 time steps; [33]), recent developments have largely overcome these challenges. Here, we employed a recurrent neural network architecture based on Gated Recurrent Units (GRUs; [34]). As our goal was to use an architecture that would learn the long-term history of the inputs, we could have used different architectures, such as based on Long Short Term Memory models [35]. The choice of a GRU-based architecture was based on the fact

that they perform well across many applications [36], as well as computational availability and expediency.

Fig 1A shows the model. Briefly, inputs $x_t$ were provided to a GRU network that produced a "hidden" representation, $h_t$, of the incoming signal based on current and past signals. For classification purposes, the hidden-layer signals were transformed to create a vector of class scores, the maximum of which determined the model's prediction of the input. Next, we describe the model formally.

Time series data from units of interest (ROIs or voxels) comprised the input vector at each time, $x_t$. The GRU network (with one or more layers) transformed its input, $x_t$, non-linearly onto its output, $h_t$, the hidden layer of the classifier system (see S1 Appendix for details). Because our goal was input classification, GRU outputs at each time, $h_t$, passed through an additionally fully-connected (FC) layer of units generating a vector of class scores:

$$\hat{y}_t = \text{Softmax}(W_{\text{FC}}h_t + b_{\text{FC}}), \tag{2}$$

where Softmax is the generalized logistic activation function. As the outputs are in the range [0, 1] and and add up to 1, they can be treated as probabilities. The output of the model was the unit label, $i$, such that it selected the class with largest probability: $\arg\max_i \hat{y}_t$. As all operations were performed at every time, $t$, the final output was a time series of label predictions. For movie clips, we employed data from 300 ROIs, so the dimensionality of the input was $N_x = 300$. The GRU network employed a single layer with 32 units ($N_h = 32$). The dimensionality of $\hat{y}_t$ was $N_y = 15$ to implement 15-way classification (based on the number of clips). For the moving-circles data, we employed data from 745 voxels ($N_x = 745$). The GRU network contained three layers of 32 units each; the third layer corresponded to the vector $h_t$ ($N_h = 32$). The dimensionality of $\hat{y}_t$ was $N_y = 2$ to implement threat vs. safe classification.

All weight matrices, $W$, required for GRU tuning (see S1 Appendix) and the bias parameters, $b$, underwent supervised training. Training sought to minimize the cross-entropy (CE) loss between predicted labels, $\hat{y}_t$, and true labels, $y_t$ (as provided by the supervisor), and was evaluated at at each time step:

$$\mathcal{L}_{\text{CE}}(\hat{y}_t, y_t; \Theta) = -\sum_{i=1}^{N_y} y_t^i \log(\hat{y}_t^i), \tag{3}$$

where the superscript $i$ indexes the dimension of the predicted and teaching signals. The set of trainable parameters was denoted as $\Theta$. The cross-entropy attains its minimum value when the two distributions, $\hat{y}_t$ and $y_t$, are identical. We defined the total loss function, $J(\Theta)$, as the average cross-entropy loss across time points (i.e., the average value across time of Eq 3), thus encouraging predicted labels to be as close to the true labels as possible across time. We minimized $J(\Theta)$ using the backpropagation through time algorithm [37]; gradient descent was optimised with the Adam optimizer [38]. The model was implemented using TensorFlow [39].

As our goal was not necessarily maximizing model performance, we explored a small set of potential architectures by varying the number of hidden layers and number of units per layer (Fig 8). For the region-based analysis, we explored the {1, 2, 3} × {16, 32, 64} space (layers by units). As accuracy was relatively high overall (>70%) and improvements were relatively modest as a function of these parameters, we chose the simplest case (one layer) but with 32 units which improved performance over 16 units (for evaluation performance, we considered all time points, even those during the first 60–90 seconds when accuracy increased sharply; see Fig 2). For the voxel-based analysis, we explored the {1, 2, 3} × {16, 32, 64, 128, 256} space.

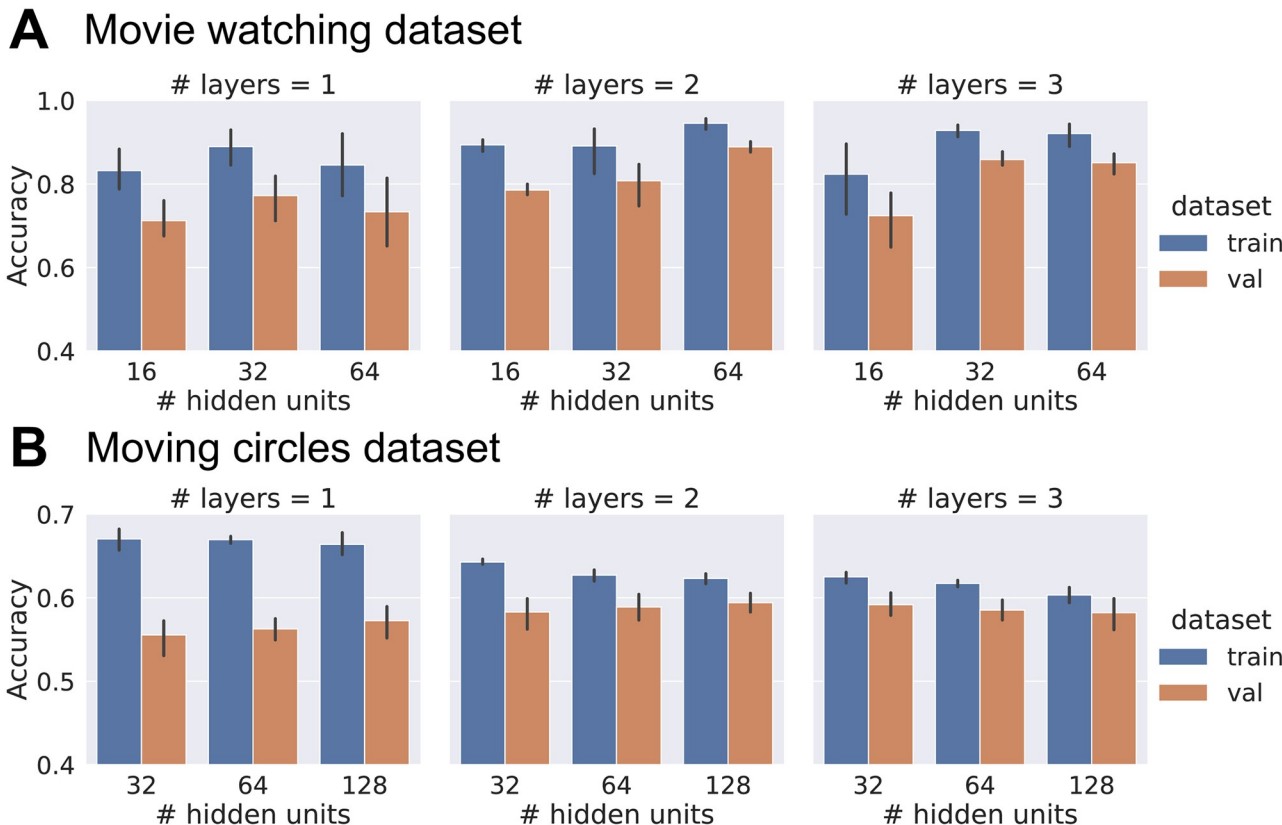

**Fig 8. Selection of model architectures via cross validation on data from 100 participants.** Val: validation portion of training data. Error bars show 95% confidence intervals across folds.

Classification of approach vs. retreat was a challenging problem, so we utilized 3 layers; we employed 32 units per layer for consistency with the clip classification architecture.

## Dimensionality reduction

The basic architecture (Fig 1A) automatically implemented considerable dimensionality reduction. For example, for movie data, the GRU layer contained 32 units, whereas inputs from 300 regions were employed. This initial step implemented a version of non-linear dimensionality reduction. However, we included a separate stage of linear dimensionality reduction that was obtained by adding an extra fully connected layer, as described next. This two-step procedures was adopted to create a modular architecture that could be probed as desired: an initial step used to learn a temporal prediction problem, and an additional dimensionality reduction stage. Our approach can thus be contrasted to employing, for instance, auto-encoder architectures with a single non-linear dimensionality reduction step associated with hidden layers [40–42].

Several dimensionality reduction techniques exist that could be used to probe lower-dimensional representations in our system. To capture the temporal relationships in the time series data in the dimensionality reduction process, we combined the GRU network with a simple additional fully connected layer for dimensionality reduction (DR-FC). We refer to this model as the *GRU encoder* (Fig 1B). Formally, GRU outputs, $h_t$, were linearly projected onto a lower-dimensional space containing units with linear activation:

$$\hat{h}_t = (W_{\mathrm{DRFC}}h_t + b_{\mathrm{DRFC}}), \tag{4}$$

where the weight matrix ($W_{\mathrm{DRFC}}$ had dimensions $N_h \times N_{\hat{h}}; N_{\hat{h}} < N_h$). We refer to the layer as the *Dimensionality Reduction Fully Connected (DR-FC)* layer. The output of this layer, $\hat{\boldsymbol{h}}_t$, is sensitive to past inputs, $\boldsymbol{x}_t$, thus effectively capturing temporal dependencies. An additional fully connected layer was used to predict labels, as in the previous subsection.

For dimensionality reduction, only the DRFC and final FC layers were trained (via standard backprogapation). In other words, the training weights of the GRU network for classification were frozen in place after that initial learning phase. For visualization, when the dimensionality of $\hat{\boldsymbol{h}}_t$ was reduced to three, we plotted "trajectories" in a state-space representation (Fig 3A).

## GRU decoder

Can low-dimensional representations obtained for classification be used to reconstruct the original data? Another GRU-based module was used to reconstruct brain activation from the learned low-dimensional representations, $\hat{\boldsymbol{h}}_t$, which we refer to as the *GRU decoder* (Fig 1C). The GRU decoder was trained independently (that is, separately from the GRU encoder), by minimizing the mean squared error between the GRU decoder output, $\hat{\boldsymbol{x}}_t$, and the original time series input, $\boldsymbol{x}_t$, at each time step.

## Saliency maps: Explaining model predictions via sensitivity analysis

Given an input activation vector, $\boldsymbol{x}_t$, the GRU classifier generates a prediction output vector, $\boldsymbol{y}_t$, encoding class-scores. As the process is supervised, the true class label, $c \in \{1, \cdots, C\}$, is known. Thus, the $c^{\mathrm{th}}$ element of the output vector, $\hat{y}_t^c$, is the class score for the input's true class label.

To help evaluate the contribution of individual input features (ROIs or voxels) to classification, we employed a saliency measure [43] that was used in the context of *sensitivity analysis*, a widely used approach in machine learning applications [44–47]. Sensitivity analysis helps in the interpretation of complex nonlinear models by representing *variation* of class-scores in terms of *variation* of individual input features. Input features are more *salient* if their class-scores are more sensitive to changes in their values.

Mathematically, the class-score, $\hat{y}_t^c$, is a multivariate function of input features, $x_t^i$:

$$\hat{y}_t^c(x_t^1, \cdots, x_t^{N_x}). \tag{5}$$

The total differential of class-score, $\mathrm{d}\hat{y}_t^c$, can be expressed as a weighted sum of the differentials of individual input features, $\mathrm{d}x_t^i, i = 1, \cdots, N_x$, weighted by their corresponding partial derivatives, $\partial \hat{y}_t^c / \partial x_t^i$:

$$
\begin{aligned}
\mathrm{d}\hat{y}_t^c &= \frac{\partial \hat{y}_t^c}{\partial x_t^1} \mathrm{d}x_t^1 + \cdots + \frac{\partial \hat{y}_t^c}{\partial x_t^{N_x}} \mathrm{d}x_t^{N_x} \\
&= \left( \frac{\partial \hat{y}_t^c}{\partial x_t^1}, \cdots, \frac{\partial \hat{y}_t^c}{\partial x_t^{N_x}} \right)^\top \left( \mathrm{d}x_t^1, \cdots, \mathrm{d}x_t^{N_x} \right) \\
&= (\nabla_{\boldsymbol{x}_t} \hat{y}_t^c)^\top \mathrm{d}\boldsymbol{x}_t \\
&\equiv \boldsymbol{w}_t^\top \mathrm{d}\boldsymbol{x}_t,
\end{aligned}
\tag{6}
$$

where the operator $\top$ is the transpose. Since the gradient vector, $(\nabla_{\boldsymbol{x}_t} \hat{y}_t^c)$, captures the sensitivity of class-score w.r.t changes in input features, we define the saliency vector, $\boldsymbol{w}_t$, as the gradient vector of the class-score evaluated at the current input, $\boldsymbol{x}_t$:

$$\boldsymbol{w}_t \equiv \nabla_{\boldsymbol{x}_t} \hat{y}_t^c. \tag{7}$$

The value of the $i^{\text{th}}$ element, $w_t^i$, is the *saliency value* for the $i^{\text{th}}$ input feature (ROI or voxel) at time $t$. The saliency indicates ROIs or voxels which, when their values change, have the largest impact on the true-class score, and consequently a larger effect on predicting the output class. The saliency vector is straightforward to implement for our model, as the gradient can be computed using the backpropagation through time algorithm [37].

To consider saliency values across participants, we z-scored the gradient values, $w_t^i$, across inputs and time steps for each participant. Because positive or negative contributions were potentially equally important (i.e., increases or decreases in brain activation altered the class prediction), we considered the absolute values of the z-scores above. Finally, we averaged these values across participants to obtain group-level saliency estimates.

To evaluate the robustness of the saliency results, we compared them to those obtained from a null model. To generate a null distribution, using test data, class labels were randomly shuffled, and saliency values computed as defined above. The procedure was repeated 1000 times, yielding a distribution of null-saliency values at each time step (see Fig 4).

## Lesion analysis

To further characterize region importance, we performed a lesion analysis. After the learning phase with the full input, particular groups of ROIs were "lesioned" (masked) by setting their input time series to zero in the test phase. Initially, we masked each network from the cortical `Schaefer-300` seven-network parcellation [32]. We then grouped ROIs within the seven "standard" networks into smaller groups using the seventeen-network `Schaefer-300` parcellation.

## Baseline models

We compared the GRU classifier to two baseline models. Training was based on computing cross-entropy loss between predicted and true labels at each time step (similar to Eq 3). The standard backpropagation algorithm was employed (Adam optimizer), as implemented in TensorFlow.

**Feed-forward network.**   The feed-forward (*FF classifier*) network consisted of three fully-connected (FC) layers. Brain inputs at time, $\boldsymbol{x}_t$, were fed into a fully-connected hidden layer, and then transformed into a vector of class scores, as with the GRU classifier. Formally, the hidden layer, $\boldsymbol{h}_t^{\text{FF}}$, and output layer, $\hat{\boldsymbol{y}}_t^{\text{FF}}$, activations were obtained as follows:

$$
\begin{aligned}
\boldsymbol{h}_t^{\text{FF}} &= \text{ReLU}(W_{\text{FC1}}\boldsymbol{x}_t + b_{\text{FC1}}), \\
\hat{\boldsymbol{y}}_t^{\text{FF}} &= \text{Softmax}(W_{\text{FC2}}\boldsymbol{h}_t^{\text{FF}} + b_{\text{FC2}}).
\end{aligned}
\tag{8}
$$

where the ReLU and Softmax operations are standard rectified linear and generalized logistic functions, respectively. The vectors $\boldsymbol{h}_t^{\text{FF}}$ and $\hat{\boldsymbol{y}}_t^{\text{FF}}$ had dimensions $N_h^{\text{FF}} = 103$ and $N_{\hat{y}}^{\text{FF}} = 15$ (for 15-way classification) respectively. The number of hidden units was set to 103 in order to maintain the total number of parameters of the classifier (32563) comparable to that of the GRU classifier (32559).

**Temporal convolutional network.**   We investigated whether static fixed-sized windows would be sufficient for modeling dynamics. A temporal convolutional network (*TCN classifier*) uses fixed-sized convolutional filters, where the output at a time depends upon a fixed temporal window, which is determined by the filter length $l$ [48, 49]. Activations after temporal convolution were then transformed to obtain class scores. Formally, brain inputs at each time, $\boldsymbol{x}_t$, were convolved temporally with $k$ filters, each with length of $l$ time steps. A single filter of size $N_x \times l$ convolved along the time dimension (input time series) and generated a scalar time

series. Therefore, $k$ such filters produced a $k$-dimensional convolutional layer activation at each time, $\boldsymbol{h}_t^{\mathrm{TCN}}$, which was used to generate class scores:

$$\hat{\boldsymbol{y}}_t^{\mathrm{TCN}} = \mathrm{Softmax}(W_{\mathrm{FC}}\boldsymbol{h}_t^{\mathrm{TCN}} + b_{\mathrm{FC}}). \tag{9}$$

The vectors $\boldsymbol{h}_t^{\mathrm{TCN}}$ and $\hat{\boldsymbol{y}}_t^{\mathrm{TCN}}$ were of dimensions $k = 25$ and $N_{\hat{y}}^{\mathrm{TCN}} = 15$, respectively. The choice of $k = 25$ filters, each of width of $l = 5$ time steps, was chosen to keep the the total number of parameters (37915) similar to that of the GRU classifier (32559).

## Predicting behavior

To predict participants' scores, we employed a GRU-based model, essentially a *GRU regressor*. Separate models were trained to predict fluid intelligence and verbal IQ [50]. In each case, all movie clips were used for training (rather than training a separate model for each clip) to promote learning representations that were not idiosyncratic to a particular clip, and thus generalizable across clips.

We applied the GRU classifier approach (Fig 1). The GRU outputs, $\boldsymbol{h}_t$ (dimensionality: $N_h = 32$), were input to a fully-connected layer with a single unit having a linear activation function. The models learned to predict a single score, $\hat{y}_t$, at every time step $t$:

$$\hat{y}_t = W_{\mathrm{FC}}\boldsymbol{h}_t + b_{\mathrm{FC}}. \tag{10}$$

Thus, we were able to generate predicted scores as a function of time, which could be compared with the true score, allowing evaluation of how model predictions evolved. We computed the mean squared error (MSE) between true, $y_t$, and predicted, $\hat{y}_t$, scores at each time point:

$$\mathcal{L}_{\mathrm{MSE}}(y_t, \hat{y}_t; \Theta) = \frac{1}{2}(y_t - \hat{y}_t)^2, \tag{11}$$

where $\Theta$ was the set of learnable parameters. The total loss was defined as the average MSE loss across all time points. The models were trained using backprogagation, using the Adam optmizer for gradient descent.

We compared our approach to connectome-based predictive modeling (CPM; [51]), possibly the state-of-the-art in this regard. In this approach, a functional connectivity matrix is initially formed for each clip based on the Pearson correlation between every pair of brain region time series. Subsequently, the entries in the matrix (or edges) that correlate with the individual differences measure beyond a set threshold (here, 0.2) are retained and a linear model is fit to predict scores based on the functional connectivity values. Given that functional connectivity is employed, CPM predictions rely on activations during the entire clip.

We tested our model statistically when making predictions at every time point. At each time $t$, once the model was trained (training set only), we evaluated chance performance based upon a null distribution obtained through permutation testing. For each iteration, we randomly shuffled the true scores of test data across participants and then calculated Spearman's rank correlation between predicted and shuffled scores (1000 iterations). The 95% confidence interval is shown by the gray zone in Fig 6. Because this test evaluates differences at every time $t$, it suffers from the problem of multiple comparisons.

To address multiple comparisons, we performed a single test (hence eschewing multiple tests) that evaluated if the *maximum* average correlation over a temporal window of length $l$ could be observed by chance. The null scenario was generated by shuffling participants (i.e., the fMRI and fluid intelligent were randomly associated, not based on participant ID). The permutation-based test draws from ideas used in the MEG/EEG field to evaluate if two time series differ in time [52]. Specifically, our method was as follows. Consider a sliding window of

**Table 1. Summary of models implemented in the paper.** Hyperparameters and other parameters: `LR`: controlled the rate of update for gradient descent; `Dr`: dropout for the GRU layer [53]; `L2`: L2 regularization coefficient for the GRU layer; `BS`: Batch Size, number of training samples before updating model parameters; `Ep`: Epochs, number of passes through the training dataset.

| Model (Dataset) | Architecture Layer (unit(s)) | # layers | LR | Dr | L2 | BS | Ep |
|---|---|---|---|---|---|---|---|
| GRU classifier (movie-watching) | 1 × GRU (32 units) | 2 | $6 \times 10^{-3}$ | $10^{-6}$ | $10^{-3}$ | 32 | 45 |
| | 1 × FC (15 units) | | | | | | |
| FF classifier (movie-watching) | 1 × FC (103 units) | 2 | $10^{-3}$ | NA | NA | 32 | 45 |
| | 1 × FC (15 units) | | | | | | |
| TCN classifier (movie-watching) | 1 × TCN (25 filters) | 2 | $10^{-3}$ | NA | NA | 32 | 50 |
| | 1 × FC (15 units) | | | | | | |
| GRU encoder (movie-watching) | 1 × GRU (32 units) | 3 | $6 \times 10^{-3}$ | $10^{-6}$ | $10^{-3}$ | 32 | 45 |
| | 1 × FC ($N_h$ units) | | | | | | |
| | 1 × FC (15 units) | | | | | | |
| GRU regressor (movie-watching) | 1 × GRU (1 unit) | 2 | $10^{-3}$ | NA | NA | 30 | 20 |
| | 1 × FC (1 unit) | | | | | | |
| GRU classifier (moving-circles) | 3 × GRU (32 units) | 4 | $6 \times 10^{-3}$ | $10^{-6}$ | $10^{-3}$ | 32 | 28 |
| | 1 × FC (1 unit) | | | | | | |

length $l$ over the entire clip time series, and compute the value of the average correlation over the sliding window. This allows us to determine the window that has the highest average correlation. For example, in Fig 6, the window of length $l = 10$ extended from $t = 139, \ldots, 148$, for fluid intelligence. To determine the probability that such value would be observed by chance, we computed the same highest average correlation based on values for shuffled participants (1000 iterations), providing a null distribution (Fig 6, S3 and S5 Figs).

The above approach handles multiple comparisons by performing a single statistical evaluation based on the *test* dataset (after training). However, it considers all temporal windows to be able to select the one that has the highest average score. Another possible approach is to select the best temporal window based on *training* data, and apply that temporal interval to the *test* data (as a test of generalization). We call this test the *oracle test* because it uses test data to guess what will work (generalize) in the test data. No correction for multiple comparisons is needed because a single test is applied to the *test* dataset. For Fluid Intelligence prediction, the best windows were 20–29 seconds for Socialnet ($p = 0.0014$) and 73–84 seconds for Starwars ($p = 0.0007$). For Verbal IQ, the best window was 128–137 seconds for Oceans (p = 0.0034). These tests pass or just miss a Bonferroni correction with 15 movies (0.05/15 = 0.0033).

## Summary of models used

Table 1 summarizes all model architectures and documents all hyperparameters used for training.

## Conclusion

In the present paper, we developed a computational approach to study and characterize spatio-temporal brain signals in the context of fMRI. The framework can be applied to other types of brain data for discovering and interpreting brain dynamics.

## Supporting information

**S1 Fig. Euclidean distances between trajectories for all movie clips.** Euclidean distances between trajectories. In the inset, the duration of every clip is indicated in parenthesis. (PDF)

**S2 Fig. Fluid intelligence predictions for all movie clips.** Fluid Intelligence predictions for all movie clips. Conventions as in Fig 6 in the main text.
(PDF)

**S3 Fig. Fluid intelligence predictions: Null distributions.** Null distributions of fluid intelligence predictions. Vertical blue lines indicate the prediction based on actual data.
(PDF)

**S4 Fig. Verbal IQ predictions for all movie clips.** Verbal IQ predictions for all movie clips. Conventions as in Fig 6 in the main text.
(PDF)

**S5 Fig. Verbal IQ predictions: Null distributions.** Verbal IQ predictions: null distributions. Vertical blue lines indicate the prediction based on actual data.
(PDF)

**S1 Video. Saliency movie for Brokovich clip.**
(MP4)

**S2 Video. Saliency movie for Star Wars clip.**
(MP4)

**S1 Appendix. Gated recurrent units.**
(PDF)

## Author Contributions

**Conceptualization:** Manasij Venkatesh, Luiz Pessoa.

**Formal analysis:** Joyneel Misra, Srinivas Govinda Surampudi, Manasij Venkatesh, Chirag Limbachia.

**Funding acquisition:** Joseph Jaja, Luiz Pessoa.

**Investigation:** Joyneel Misra, Srinivas Govinda Surampudi, Manasij Venkatesh, Chirag Limbachia, Luiz Pessoa.

**Methodology:** Joyneel Misra, Srinivas Govinda Surampudi, Manasij Venkatesh, Chirag Limbachia.

**Project administration:** Luiz Pessoa.

**Resources:** Joseph Jaja, Luiz Pessoa.

**Software:** Joyneel Misra, Srinivas Govinda Surampudi, Manasij Venkatesh, Chirag Limbachia.

**Supervision:** Joseph Jaja, Luiz Pessoa.

**Writing – original draft:** Joyneel Misra, Srinivas Govinda Surampudi, Manasij Venkatesh, Luiz Pessoa.

**Writing – review & editing:** Joyneel Misra, Srinivas Govinda Surampudi, Joseph Jaja, Luiz Pessoa.

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
