## [Decision Letter · Decision Letter 0]

26 May 2021

Dear Dr. Pessoa,

Thank you very much for submitting your manuscript "Learning brain dynamics for decoding and predicting individual differences" for consideration at PLOS Computational Biology.

As with all papers reviewed by the journal, your manuscript was reviewed by members of the editorial board and by several independent reviewers. Your paper was in general well received, but several issues, some of them serious, were raised. In light of the reviews (below this email), we would like to invite the resubmission of a significantly-revised version that takes into account the reviewers' comments.

We cannot make any decision about publication until we have seen the revised manuscript and your response to the reviewers' comments. Your revised manuscript is also likely to be sent to reviewers for further evaluation.

Sincerely,

Daniele Marinazzo

Deputy Editor

PLOS Computational Biology

Daniele Marinazzo

Deputy Editor

PLOS Computational Biology

Reviewer's Responses to Questions

**Comments to the Authors:**

Reviewer #1: Thank you for inviting me to review this manuscript by Pessoa and colleagues, in which the authors train a GRU with functional MRI data in order to classify different movie clips and task states, while also retaining the ability to track low-dimensional trajectories over time. Overall, I found the study to be of interest, however there were a number of sections that I believe would benefit from more explicit justification and expansion.

• The authors may wish to provide some insights into why a GRU architecture was chosen over other potential options (e.g., vanilla feed-forward networks; convNets; LSTMs; etc). I appreciate that they used two of these as benchmark networks, however it would benefit the reader if the reasons for including them (or not) were made more explicit.

• I apologise if I missed something, but how does training a GRU with an extra fully-connected hidden layer lead to low-dimensional representations of the data? Is it simply due to the fact that the added hidden layer has a smaller number of nodes? And if so, how did the authors determine precisely how low-dimensional to make this hidden layer?

• Fig 3. The results of Figure 3 are mildly surprising. Were the authors not surprised that their different test benchmarks performed so poorly, particularly when the only major differences between GRUs and ANNs/ConvNets is related to local nodal recurrence. It was also slightly strange that each of the null networks was strongly above chance (i.e., greater than 8%) but also hovered around the same value (~40%). What is the significance of this value?

• Fig 4. Can the authors provide any intuition as to why the trajectories of the different movie clips arise from the origin of the 3D plot?

The authors interest in fluid intelligence and fMRI may be augmented by reference to both https://doi.org/10.1038/nn.4135 and https://doi.org/10.1038/s41593-018-0312-0

Reviewer #2: In this work, Pessoa et al. test a neural network architecture to decode fMRI-based brain dynamics and classify brain patterns elicited by movie-watching and a continuous experimental paradigm. They explore the associations between the employed GRU network and dimensionality reduction, behavior and brain activation through salience maps. The manuscript is generally clearly written and logically implemented. However, some of the characteristics of this methodology should be more clearly explained and motivated, before I can recommend this work for publication. Please find my comments below.

Major comments:

- One of the goals of using this approach was to exploit the temporal richness of the fMRI brain signals. However, as the authors acknowledge, "temporal shuffling of the time series still allowed the GRU classifier to attain classification at around 50% correct". In other words, breaking temporal information still gives ~50% accuracy after the GRU modeling procedure. Isn't that worrisome? As a "sanity" check, could the authors include a resting-state condition in the data pool, to see how it will affect the classification results?

- If I understood Fig. 4 correctly (I'd recoomend to improve the caption of 4C), the trajectories in 4A) are based on the first 3 PCA components of the GRU outputs h_t, which already allow for a high classification accuracy (~60%). However, these very same 3 components have very little to do with the original brain signal ( they only explain ~2 % of the variance of the original data!). This is somewhat in line with the common viewpoint of considering neural networks architectures as "black box operators", that output complicated non-linear projections of original signals, which are then harder to trace back. Therefore, I don't understand how informative are the trajectories in 4A), since they have very little to do with the original brain signal, and are only related to the specific GRU architecture. This point should be better elaborated in the manuscript, to warn the reader on potential misleading interpretations of these NN classifiers.

- On a related note, this also makes me wonder: how accurate is the approximation in equation 5), on which the brain salience maps were based?

- Similarly, what is the advantage of this complicated GRU-based model for the voxel prediction maps depicted in Fig. 7 with respect to, for instance, a simple GLM-based (SPM style) analysis? Wouldn't the GLM approach give you the same brain maps shown in Fig. 7?

- I know that in our community there is the increasing need of connecting new computational models with behavior and/or clinical scores. However, I really have trouble understanding

why the way a person watches Star Wars or Home Alone should be predictive of that person's "fluid intelligence". The current manuscript lacks a proper discussion of these results.

I would appreciate a more in-depth reasoning on these "neuroscientific" findings, rooted in the state-of the art of naturalistic task analysis. In other words, it is clear that the GRU black box is very effective at classifying different naturalistic tasks, but what do we learn about the brain that we did not know before?

Minor comments:

- Methods- Was band pass filtering applied to the fMRI data? What about Global signal regression? Would those impact the classification results?

- Methods. Moving-circles data: how were the voxels selected?

- Behavior. "Because of multiple comparisons across time, we performed an additional test considering all time points simultaneously." It is not clear how the authors correct for multiple comparisons. Please clarify.

Reviewer #3: The paper presents a spatiotemporal analysis framework for dynamic classification ,regression and visualization of fMRI data. The authors develop a deep learning architecture based on GRUs (a recurrent neural net architecture) using three related components: a classifier to predict the class of stimuli, an encoder for dimensionality reduction to perform visualization and a decoder. They also present saliency maps which enable an interpretability of the model and provide information on the importance of different brain regions / voxels to the classification. This is an advantage of the approach compared to other nonlinear techniques which cannot map back to the original input space.

I enjoyed reading the paper and believe it is an important contribution of interest to the readers of PLOS Computational Biology.

Major comments

1. The authors should comment in the paper why they choose to train three separate networks instead of training an end-to-end neural network for classification, encoding and decoding, i.e. an alternative architecture could have been a jointly trained autoencoder and classifier but perhaps this would be less stable to train or result in poorer performance. In addition regarding the encoder, since it shares the GRU with the classifier (these aren’t actually fully separate), this should be made more explicit in the figure.

2. Related work: the introduction should also briefly discuss how this paper relates to recent work on dynamical functional connectivity.

3. The authors make the point that decoding is typically static, based on for example reducing the activity to a functional connectivity matrix or averaged activity. A static method should be added as a benchmark to the classification task, similar to including CPM as a static baseline for predicting fluid intelligence and verbal IQ.

4. For the TCN classifier, increasing the length of the kernel from 5 to 10 time steps improved performance. Does a further increase to larger kernel widths (20?40?) further improve classification accuracy? This also relates to the author’s discussion on spatiotemporal representations that long-term dependencies are essential.

5. Regarding the low-dimensional trajectories as well as trajectory distances, what can be learned from this visualization, i.e. what is the aim of visualization? Is the organization of video clips meaningful in any way?? The result that classifying on the low dimensional space (3-10 dimensions) yields classification accuracy close to the full dimension is interesting as it indicates the intrinsic dimensionality of the data.

Minor comments:

• Can the authors comment on whether saliency maps can be generated for behavior prediction (regression task), as done for the classification task.

• Line 12 “to infer mental processes and processes from brain signals.” Missing word?

• GRU – main text should mention detailed explanation is in supplementary

• Line 131 N_y should be N_h

• Line 148 – should be layers by units

• Line 256 – how was the threshold of 0.2 chosen?

• Table 1 – add number of layers

• Line 284 - “were obtained with not used” – missing word?

• Red line missing in Fig S4 for flower clip

Reviewer #4: The authors propose using modern recurrent neural networks (in this case, a network with gated recurrent units; GRUs) to analyze spatial-temporal brain data in rich, continuous experimental settings like movie watching. Considering most fMRI work is based on static images, this seems like a potentially interesting approach to reveal what previous work could not.

Overall, the authors did a lot of work, but did not sufficiently motivate and explain the methods in a way that convinced me that we learnt much new - or what the new part was. The authors need to explain and motivate their approach, what their findings are, and why this is the way to go (at least for this kind of data). Below, I note major and minor points that I hope will be helpful to improve the manuscript.

Major concerns

1. General point on motivation and introducing the method. It was not clear to me what we benefit from this analysis from a broad perspective, or from the specific analysis steps. There is a brief general note on what they will do and why (e.g. decode, reduce dimensionality), but not clear what the conceptual significance is (test what kind of information there is some brain area, explain what temporal information is capture and why it is interesting).

2. Interpretation problem from nonlinear decoders. What can we say from a non-linear decoder?

Standard MVPA is linear - and the point is if you can read out information from a certain brain area from a linear decoder, the content is there in an explicit way. For example, you can decode anything from the retina if you use a non-linear decoder, but that doesn't mean everything is represented there. For example, see Tong and Pratte 2012, p487, DiCarlo et al., 2012 on tangled representations, Kriegeskorte & Kreiman 2011.

3. What does the decoder tell us? What did we learn?

The classifying decode which movie clip it is across participants. This tells us the brain holds some info about movie clip. Is this the conclusion? What have we learnt? Is it based on visual, auditory information, or semantic content?

More generally: what would decoding of a movie clip mean - we probably don't care that area X represents encodes movies, or distinguishes different movies… What is the important point?

Although the authors argue that the method give spatial and temporal insights, there doesn't seem to be any spatial or temporal interpretation since it uses all the data to construct the classifier. Training on temporal data then testing on specific time points - does this give any info on that specific time point, is there a problem with the classifier having inputs/info from other timepoint?

The authors use saliency maps for finding interesting spatial or temporal points, but it's unclear if this is meaningful (see point below on saliency maps).

4. Why GRUs, why is it good for the current setting? Lack of a motivation/explanation

Typically, GRUs are used to predict the next time step. Here it seems like it tries to do that too, but then prediction is then on a 15-way classifier to classifier movie clips across participants. Not clear how the temporal prediction, applied to across subjects analysis and not predicting time, helps. Authors need to clarify this - and maybe it is an important method point that I missed.

5. Saliency maps - the paper authors cite was for images in convnets, a visualizing technique based on images which usually finds where the 'main object' is. Is this transferrable to brain data? What is a 'salient' activation/timepoint in the brain in this case?

Input feature is time and/or space - what would a salient timepoint mean? Strong activation, or strong decoder? Is it just a (statistical) way to extract the strong areas/timepoints or something more? Is it related to saliency in the clips (probably not?)?

Not much details in the results and discussion. Would be good to have this from the methods and to fully motivate the meaning and purpose of this measure.

6. Dimensionality reduction - this is non-linear, so again, this seems like there might be a problem for interpretation, especially when it is used to test if you can decode from the low-dimensional space - i.e. same problem as a non-linear decoder above. Furthermore, this is now a non-linear space - which could be high dimensional in linear space?

Separately - dimensions increased variance explained - is it still a low-d manifold? Looks like it keeps increasing. That the first few dimensions take much of the variance is common (e.g. in PCA), and here it's non-linear which might mean it's more. This is not so much an issue of method but interpretation.

Maybe more important: it is 10d with time - is this not much higher dimension than 10?

7. Predicting 'behavior' (e.g. IQ scores) - why is this done over time? Is it actually predicting a single score over time? Why would you predict something multiple times when there is only 1 score (stats point) - it seems more valid (and powerful) to predict one score per participants with all the data. More importantly, why would you theoretically predict that it varies, and correlates with movie watching?

Minor

8. Discussion is very short and basic, reiterating the results without much interpretation of what they mean. If the contribution is more about a method, more discussion should go into the methods and comparison with previous methods. If it's about prediction (in the ML sense), then they can also be more clear about that, without interpretation.

9. The GRU model does better than other models that don't take into account time. These are some good comparisons. However, it was unclear to me if the GRU was therefore more complex (it is in the sense that it takes into account temporal info, so maybe that's unavoidable) so did better, and if there are more parameters (which would be unfair).

10. "As our goal was not necessarily maximizing model performance, we explored a small set of potential architectures by varying the number of hidden layers and number of units per layer "

If it's not, what's the purpose of testing multiple models? Having multiple models show similar effects could be informative to show that it's reliable across different model settings, but it seems like the authors did choose the best model.

Why in the HCP 1 layer enough and in the moving circles task 3 layers needed? Classification task was harder, probably less information (e.g. visual). But this is now more difficult to interpret. What does this tell us - does it tell us anything apart from decoding is harder because there less information? Then making the model more complex you can decode more - this is obvious, but not clear how it's interpretable neurally.

11. Train on subset of participants, test on others (100 train, 76 test). This is fine but why these numbers? Worth justifying in the text (e.g. more training data can be more desirable than split half).

12. Threat vs safe - it seems like the brain region would have come up in a more traditional univariate and/or multivariate (e.g. linear SVM) analysis as well. What is added here?

**Have the authors made all data and (if applicable) computational code underlying the findings in their manuscript fully available?**

Reviewer #1: None

Reviewer #2: Yes

Reviewer #3: Yes

Reviewer #4: **No: **"All data and code will be made available upon publication." I can see the GitHub repo but not the processed data. Not sure if they will have a repo for that or will just point to the HCP?

PLOS authors have the option to publish the peer review history of their article (what does this mean?). If published, this will include your full peer review and any attached files.

Reviewer #1: No

Reviewer #2: No

Reviewer #3: No

Reviewer #4: No
---

## [Decision Letter · Decision Letter 1]

19 Aug 2021

Dear Mr Misra,

We are pleased to inform you that your manuscript 'Learning brain dynamics for decoding and predicting individual differences' has been provisionally accepted for publication in PLOS Computational Biology.

Best regards,

Daniele Marinazzo

Deputy Editor

PLOS Computational Biology

Daniele Marinazzo

Deputy Editor

PLOS Computational Biology

Reviewer's Responses to Questions

**Comments to the Authors: **

Reviewer #1: The authors have adequately addressed my concerns.

Reviewer #2: The authors have addressed all my concerns.

Reviewer #4: The authors have made many clarifications and rephrasings that better expresses the purposes of the paper and its reach.

The lesion results were very nice. One idea for an additional analysis is if you lesion brain regions across 2 (or more) salient networks – do they cause an additive amount of decrease in accuracy (because they are different, independent contributions, e.g. 10% drop + 10% drop in accuracy = 20%) or not (because they are not independent – e.g. visual-motor systems might be representing visual actions, and so you get still a 10% drop). However, it might not be necessary here. It might be worth discussing / doing in the future.

Individual differences – OK, makes sense with anxiety and suspense. Right - with IQ, I guess one reason could be to test which timepoints might relate to IQ (e.g. complex storyline / whether participants understood something). With the results here I think it’s probably less interesting (or difficult to find out) but proposing it as a method makes sense.

General comment: A suggestion to the authors for replying to reviews (a suggestion to me from my advisors) – include the points / changes to the manuscript in the reply verbatim (e.g. in quotation marks). It was inconvenient to go to the referenced lines (which is better than no reference), but difficult to figure out how the points were addressed when there were no references.

**Have the authors made all data and (if applicable) computational code underlying the findings in their manuscript fully available?**

Reviewer #1: None

Reviewer #2: Yes

Reviewer #4: Yes

PLOS authors have the option to publish the peer review history of their article (what does this mean?). If published, this will include your full peer review and any attached files.

Reviewer #1: No

Reviewer #2: No

Reviewer #4: No

---

## [Editor Report · Acceptance letter]

30 Aug 2021

PCOMPBIOL-D-21-00636R1 

Learning brain dynamics for decoding and predicting individual differences

Dear Dr Misra,

I am pleased to inform you that your manuscript has been formally accepted for publication in PLOS Computational Biology. Your manuscript is now with our production department and you will be notified of the publication date in due course.

With kind regards,

Andrea Szabo
